# Breast Tissue Restoration after the Partial Mastectomy Using Polycaprolactone Scaffold

**DOI:** 10.3390/polym14183817

**Published:** 2022-09-13

**Authors:** Seung-Jun Jwa, Jong-Min Won, Do-Hyun Kim, Ki-Bum Kim, Jung-Bok Lee, Min Heo, Kyu-Sik Shim, Han-Saem Jo, Won-Jai Lee, Tai-Suk Roh, Woo-Yeol Baek

**Affiliations:** 1Department of Plastic and Reconstructive Surgery, Severance Hospital, Yonsei University College of Medicine, Seoul 03722, Korea; 2PLCOskin Co., Ltd., Seoul 120-752, Korea; 3Department of Biological Science, Sookmyung Women’s University, Cheongpa-ro 47-gil 100, Yongsan-gu, Seoul 04310, Korea; 4Department of Plastic and Reconstructive Surgery, Institute for Human Tissue Restoration, Severance Hospital, Yonsei University College of Medicine, Seoul 03722, Korea

**Keywords:** polycaprolactone, partial mastectomy, tissue restoration, breast restoration

## Abstract

As breast conserving surgery increases in the surgical treatment of breast cancer, partial mastectomy is also increasing. Polycaprolactone (PCL) is a polymer that is used as an artifact in various parts of the human body based on the biocompatibility and mechanical properties of PCL. Here, we hypothesized that a PCL scaffold can be utilized for the restoration of breast tissue after a partial mastectomy. To demonstrate the hypothesis, a PCL scaffold was fabricated by 3D printing and three types of spherical PCL scaffold including PCL scaffold, PCL scaffold with collagen, and the PCL scaffold with breast tissue fragment were implanted in the rat breast defect model. After 6 months of implantation, the restoration of breast tissue was observed in the PCL scaffold and the expression of collagen in the PCL scaffold with collagen was seen. The expression of TNF-α was significantly increased in the PCL scaffold, but the expression of IL-6 showed no significant difference in all groups. Through this, it showed the possibility of using it as a method to conveniently repair tissue defects after partial mastectomy of the human body.

## 1. Introduction

Mastectomy for breast cancer started with radical surgery [1], and following several studies, the procedure of choice for early Stage I and II breast cancer was modified radical mastectomy [2] and then developed into breast conserving surgery, such as quadrantectomy and partial mastectomy [3]. It is estimated that about 1.3 million people worldwide undergo breast conserving surgery every year [4]. A recent study found that up to 80% of breast cancers were safely treated with breast conserving surgery (BCS) [5]. With advances in breast cancer diagnostic technology, many clinical studies, and advances in chemotherapy and radiation therapy, the current goal of surgical resection is as minimal as possible. Oncoplastic surgery such as local tissue rearrangement, mastopexy approach, and reduction mammoplasty has advantages of manipulating autologous breast tissue [6]. However, since normal breast tissue must be excised, the volume does not increase, and the possibility of cancer recurrence cannot be completely excluded. In order to overcome this, there is a ‘delayed-immediate’ method to reconstruct before receiving radiation therapy after confirming that it is finally pathologically clear after partial mastectomy [7], but the fact that it is a two-stage procedure can act as a disadvantage. In addition, in order to obtain a cosmetically satisfactory result after breast conserving surgery, there are various factors to consider, such as tumor:breast volume percent or the tumor size and location [4].

Polycaprolactone (PCL) is a biodegradable polymer that is approved by the Food and Drug administration (FDA) on specific applications in the human body such as a bone defects [8] and cartilage tissue [9]. PCL scaffolds have the potential to be used for soft tissue reconstruction because of its low mechanical strength [10]. A 3D printed polycaprolactone scaffold showed promising potential of lipogenesis in vitro [11]. Here, we hypothesized that a PCL scaffold can be utilized for the restoration of breast tissue after a partial mastectomy. To demonstrate the hypothesis, a sphere-type PCL scaffold was manufactured by 3D printing and three types of PCL scaffold were implanted in the rat breast defect model.

## 2. Materials and Methods

### 2.1. Fabrication of PCL Scaffold for Patial Mastectomy

The PCL scaffold in the present study was fabricated by a 3D printing procedure as shown in Figure 1A. To manufacture the PCL scaffold, a PCL filament (Medical grade) was purchased from Evonik Ltd., Essen, Germany. The sphere-type PCL scaffold was designed by Fusion 360 computer-aided design (CAD) software (Autodesk, Inc., San Rafael, CA, USA). The CAD model was exported to an STL file and imported to ideaMaker 3D print slicer software (Raise3D, Irvine, CA, USA), where printing parameters were set and the structure was sliced into 0.2 mm sections. The file was exported to G-code file and printed using pro2 3D printer (Raise3D, Irvine, CA, USA) that was fitted with a 0.40 mm nozzle. The PCL scaffold was printed as 10 mm diameter at 130 °C and a speed of 10 mm/s. The PCL scaffold was sterilized by irradiation with 25 kGy of E-beam in EB Tech, Daejeon, Korea and examined for further in vivo experiments. To evaluate the surface structure of the PCL scaffold, scanning electron microscope (SEM) 200× magnification was analyzed using a field emission scanning electron microscope (Merlin, Carl Zeiss, Oberkochen, Germany).

### 2.2. Mechanical Property of PCL Scaffold for Patial Mastectomy

To determine the mechanical properties of the PCL scaffold, a compression test was preformed using a universal testing machine (TESTONE CO., LTD., Siheung, Korea). The compression responses of the PCL scaffold during loading were recorded. The load-displacement measurements using a 50 kgf load cell at a constant crosshead speed of 5 mm/min. There were three samples (*n* = 3) that were used for compression test. To test the recovery capability of the PCL scaffold before and after sterilization, compression tests were performed. The samples were compressed at a rate of 5 mm/min up to 25%, 50%, 75%, and maximum compression of their initial height. The loading and unloading states were performed for 5 min respectively.

### 2.3. PCL Scaffold Implantation after Patial Mastectomy in Rat Model

The experiment in the present study was performed as shown in Figure 1B. All of the animal experiments in the present study were approved by Institutional Animal Care and Use Committee (IACUC) of Yonsei University College of Medicine (Approval Number: 2021-0019). A total of 21 8-week-old female Sprague–Dawley rats were provided by ORIENT BIO, Seongnam, Korea. To investigate the tissue restoration capacity of the PCL scaffold and the synergistic effects of the PCL scaffold with extracellular matrix and autologous tissue implantation after partial mastectomy. There were three types of PCL scaffold that were examined: (1) PCL scaffold only (PCL); (2) PCL scaffold with injection of 1 mL of 3% collagen Type 1 solution within the PCL scaffold (MSbio, Inc., Seongnam, Korea) (PCL-col); and (3) PCL scaffold with autologous mammary gland fragment that was produced by surgical procedure immediately (PCL-tissue). The rats were anesthetized with 24 mg/kg of Alfaxan (Jurox, Rutherford, Australia) intraperitoneally and 2% of Isoflurane (HANA PHARM Co, Ltd. Seoul, Korea) respiratory during implantation surgery. Approximately 0.5 cm^3^ of mammary gland was surgically incised in two sites (left and right) of each rat breast and one type of scaffold was implanted in the same region.

To evaluate the time-dependent soft tissue restoration within the PCL scaffold, pre-warmed 2 mL of Visipaque (GE Healthcare, Chicago, IL, USA) was intravenously injected and analyzed with Quantum GX2 microCT Imaging System (PerkinElmer, Waltham, MA, USA) in 4 months and 6 months after surgery.

### 2.4. Histological Analysis

To investigate the histological analysis, animals were sacrificed using CO_2_ gas in 6 months after surgery. All of the groups in the present study were surgically isolated. The samples were washed with 2% of antibiotics-antimycotics (Thermo Fisher Scientific, Waltham, MA, USA) in phosphate buffered saline (PBS) (Welgene, Seoul, Korea) and fixed with 4% paraformaldehyde (Junsei, Tokyo, Japan) at 4 °C overnight. The PCL scaffold-implanted samples were cut into half to analyze the tissue within the scaffold and all of the samples were embedded in paraffin. For the histological analysis, Hematoxylin and Eosin (H&E) staining and Masson’s trichrome (MT) staining were performed. The image was captured with a BX43 microscope (Olympus, Tokyo, Japan).

### 2.5. Evaluation of Soft Tissue Restoration within PCL Scaffold

To confirm the characteristics of the tissue inside the PCL scaffold, immunofluorescence (IF) was performed. The section was deparaffinized with xylene (Ducksan, Seoul, Korea) and hydrated using 100% to 50% ethanol (Ducksan, Seoul, Korea) and deionized water. Antigen retrieval was performed followed by heat-induced epitope retrieval (HIER) using citrate buffer (pH 6) (Sigma, St. Louis, MO, USA) that was supplemented with 0.05% Tween 20 (Junsei, Tokyo, Japan) on microwave for 20 min. The blocking of unspecified marker in these samples was performed using VECTASTAIN^®^ Elite^®^ ABC Kit (VECTOR laboratories, Burlingame, CA, USA) as described by the protocol. Primary antibodies including mouse anti-collagen 1 antibody (1:200) (Abcam, Cambridge, UK) and rabbit anti-perilipin-1 antibody (1:200) (Abcam, Cambridge, UK) were treated at 4 °C overnight. After washing with 0.5% Tween 20 in PBS (PBST), the secondary antibodies including Alexa 488-conjugated anti-mouse IgG (1:200) antibody (Abcam, Cambridge, UK) and Alexa 555-conjugated anti-rabbit IgG (1:200) antibody (Abcam, Cambridge, UK) were treated at room temperature for 2 h. The nuclei were counterstained with 100 μM of hoechst 33342 (Thermo Fisher Scientific, MA, USA) at room temperature for 5 min. After staining, the samples were mounted with ProLong™ Gold Antifade reagent (Invitrogen, Carlsbad, CA, USA). The image was captured using a BX53 fluorescence microscope (Olympus, Tokyo, Japan) and the expression of fluorescence was measured using ImageJ Fiji software (version 2.35, National Institutes of Health, Bethesda, MD, USA).

### 2.6. Evaluation of Inflammation after PCL Scaffold Implantation

To evaluate the biocompatibilities of the PCL scaffold after partial mastectomy, the level of inflammation was confirmed by immunohistochemistry (IHC). The samples were routinely deparaffinized with xylene and hydrated with graded ethanol and deionized water. After antigen retrieval, 3% H_2_O_2_ in methanol (Ducksan, Seoul, Korea) was treated for 10 min for peroxidase inactivation. Further process was followed by VECTASTAIN^®^ Elite^®^ ABC Kit (VECTOR laboratories, Burlingame, CA, USA) as described by the protocol. Briefly, blocking buffer was treated at room temperature for 20 min. Primary anti-tumor necrosis factor alpha (TNF-α) and anti-interleukin 6 (IL-6) mouse antibodies (Abcam, Cambridge, UK) were treated to each slide at 4 °C overnight. After washing with PBST, secondary biotinylated antibody was treated at room temperature for 2 h and VECTASTAIN Elite ABC Reagent was treated for 30 min. These samples were stained with 3,3′-diaminobenzidine (DAB) peroxidase substrate solution (VECTOR laboratories, Burlingame, CA, USA) for 2 min. The nuclei were counterstained with Mayer’s hematoxylin solution (Abcam, Cambridge, UK) and treated with 0.1% sodium bicarbonate. After staining, these samples were dehydrated with graded ethanol and xylene and mounted with Eukitt Quick-hardening mounting medium (Sigma, St. Louis, MO, USA). The image of the samples was captured using a BX43 microscope (Olympus, Tokyo, Japan) and the colorization of the samples were measured using ImageJ Fiji software (version 2.35, National Institutes of Health, Bethesda, MD, USA).

### 2.7. Alizarin Red S Staining after PCL Scaffold Implantation

To evaluate the microcalcification after PCL scaffold implantation, all the samples were routinely deparaffinized and hydrated. After hydration, the samples were incubated in 2% Alizarin Red S (Sigma, St. Louis, MO, USA) in deionized water (pH 4) for 10 min. After washing, the samples were dehydrated with graded ethanol and xylene. The samples were mounted with Eukitt Quick-hardening mounting medium (Sigma, St. Louis, MO, USA) and images of the samples were taken using a BX43 microscope (Olympus, Tokyo, Japan) and the colorization of staining was measured using ImageJ Fiji software (version 2.35, National Institutes of Health, Bethesda, MD, USA).

### 2.8. Statistical Analysis 

All of the statistical analyses in the present study were visualized and analyzed using GraphPad Prism 5 software (GraphPad Software Inc., San Diego, CA, USA). The data are represented as the mean ± SEM. The volume quantification of soft tissue within the PCL scaffold that was measured by microCT was evaluated by two-way ANOVA analysis (** *p* < 0.01, *** *p* < 0.001). A significant difference of immunofluorescence and immunohistochemistry was analyzed by one-way ANOVA analysis (* *p* < 0.05, ** *p* < 0.01, *** *p* < 0.001).

## 3. Results

### 3.1. Fabrication and Mechanical Properties of PCL Scaffold

The PCL scaffold was designed as a sphere and successfully fabricated that had a diameter of 10 mm (Figure 1A). The PCL scaffold has a porous structure, thereby increasing its flexibility. To evaluate the surface structure of the PCL scaffold, an SEM image was obtained (Figure 1B). The SEM image showed that the PCL scaffold has a multi-layer structure to sustain the sphere structure and the PCL scaffold is well-defined without nanoscale pore distribution. To evaluate the mechanical properties of the PCL scaffold, the compressive strength of the PCL scaffold was measured (Figure 1C). The PCL scaffold has 8.3 kgf compressive strength but the compressive strength of the PCL scaffold was decreased to 4.6 kgf after E-beam sterilization, respectively. In addition, to measure shape recovery ability of the PCL scaffold, compression tests were performed (Figure 1D). The shape recovery ability of non-sterilized PCL scaffolds at 25%, 50%, 75%, and maximal compression were 89.5 ± 0.7%, 81.8 ± 0.5%, 73.0 ± 1.6%, and 63.6 ± 2.2%, respectively. On the other hand, the shape recovery ability of the sterilized PCL scaffolds at 25%, 50%, 75%, and maximal compression were 88.5 ± 1.8%, 80.3 ± 1.4%, 69.8 ± 1.6%, and 65.4 ± 2.3%, respectively. Thus, the compressive strength of the PCL scaffold was somewhat decreased after sterilization but the morphological recovery ability was maintained.

### 3.2. Time-Dependent Tissue Restoration after PCL Scaffold Implantation

Next, to investigate the time-dependent tissue restoration after PCL scaffold implantation in partial mastectomy, a microCT imaging system was used (Figure 2A). As a result, the tissues were fully restored in all of the PCL scaffold-implanted groups after 2 months (data not shown). In addition, it was determined that at least two different types of tissues are restored within the PCL scaffold. To assess the soft tissue restoration after PCL scaffold implantation, soft tissue was visualized and the volume of soft tissue that was measured depends on the density of tissue (Figure 2B,C). In the present data, an amount of 101.6 ± 21.2 mm^3^ of soft tissue was measured in the PCL group in 4 months and 178.5 ± 1.5 mm^3^ of soft tissue was measured in 6 months. It was determined that 187.6 ± 9.3 mm^3^ of soft tissue was measured in the PCL-col group in 4 months but 146.8 ± 1.7 mm^3^ of soft tissue in 6 months. In PCL-tissue group, 56.0 ± 7.4 mm^3^ of soft tissue was measured in 4 months and 104.4 ± 9.7 mm^3^ of soft tissue was measured in 6 months. Thus, the implantation of PCL scaffold restored soft tissue in partial mastectomy.

### 3.3. Histological Analysis after PCL Scaffold Implantation

Moreover, the soft tissue that was restored within PCL scaffold was characterized by histological analysis. After microCT imaging in 6 months, the animals were sacrificed and operated on where the skin, breast, and implanted scaffolds were obtained. The data of H&E and MT showed that the fat tissue is invasively restored in the PCL group (Figure 3A). On the other hand, fibrous tissue was restored in the PCL-col group and a little amount of fat tissue was restored in the PCL-col group relatively. In the PCL-tissue group, co-implanted breast tissue fragments including fat, lymph node, and mammary gland was observed in the PCL-tissue group and the fragment was enveloped with fibrous tissue.

To characterize the restored tissue within the PCL scaffold, immunofluorescence was performed (Figure 3B,C). As a result, the intensity of collagen 1 was lowest in the PCL group but the intensity of perilipin-1 was highest in the PCL group compared with the PCL-col and PCL-tissue group relatively. In detail, the intensity of collagen 1 in the PCL-col group and PCL-tissue group was 1.6-fold higher than the PCL group. On the other hand, the intensity of perilipin-1 in the PCL-col group was 5.5-fold lower than the PCL group and the intensity of perilipin-1 in the PCL-tissue group was 2.1-fold lower than the PCL group. Thus, it was determined that fat tissue was restored within the PCL scaffold and fibrous tissue was restored by collagen and breast tissue fragment co-implantation.

### 3.4. Inflammatory Properties of PCL Scaffold

Additionally, to confirm the immune reaction after PCL scaffold implantation, immunohistochemistry was performed (Figure 4A,B). As a result, the expression of TNF-α was significantly higher in the PCL group and statistical difference was not detected in the PCL-col group and PCL-tissue group compared to the sham group. In addition, there was no significant difference in the expression of IL-6 in all of the groups.

Microcalcification is one of the well-known foreign body reactions and features in early breast cancer. Accordingly, microcalcification after PCL scaffold implantation was confirmed by Alizarin Red S staining (Figure 4C,D). As a result, it was determined that there was significant increase in the PCL group but there was no significant difference in the PCL-col group and the PCL-tissue group compared to the sham group. Thus, the immune reaction was induced by PCL scaffold implantation but regulated by collagen and breast fragment co-implantation.

## 4. Discussion

The aim of this study is to verify the ability of a PCL scaffold for breast tissue regeneration. The PCL scaffold which has porous structure provides an excellent environment for cells to promote adhesion, proliferation, and differentiation while filling the breast tissue [12]. Since most of the breast tissue is composed of adipose tissue, the tissue we need to reconstruct must also have a high proportion of adipose tissue. In the present study, we sought to demonstrate that the implantation of a PCL scaffold increases the growth of adipose tissue. It is observed that adipose tissue was time-dependently restored by PCL scaffold implantation after partial mastectomy in rat by microCT analysis. However, soft tissue was observed after implantation of the PCL scaffold with collagen, but the volume of soft tissue was gradually decreased. It is expected that the initial collagen is reassembled into relatively denser fibrous tissue compared to adipose tissue. The implantation of the PCL scaffold with rat breast tissue fragments showed the lowest volume of soft tissue restoration.

Therefore, it is expected that the PCL scaffold is effective in the breast restoration after partial mastectomy. However, it is also revealed that implantation of a PCL scaffold increases pro-inflammatory cytokine and microcalcification, relatively. Many studies have been conducted to decrease the inflammation by modification of the PCL scaffold [13,14]. Likewise, it is needed to increase the anti-inflammatory properties for further studies. On the other hand, the implantation of the PCL scaffold with collagen increased the fibrous tissue and decreased the inflammation properties compared with the implantation of the PCL scaffold so that the PCL scaffold with collagen has the potential to be utilized for fibrous connective tissue regeneration such as bone [15,16,17] and cartilage [18,19], relatively.

The PCL scaffold was manufactured with a 10 mm diameter in the present study, which is sufficient to repair the defect by partial mastectomy in human breast. However, the ratio of volume of the PCL scaffold to rat breast is by no means small. Therefore, further studies for PCL scaffolds for partial mastectomy may be designed with the higher animal model to apply to the clinical approaches. In addition, the mechanical strength of the PCL scaffold was stiffer than the glandular tissue (2~66 kPa) and adipose tissue (0.5~25 kPa) in human breast [20]. It is necessary to reduce the mechanical strength and increase the flexibility of the PCL scaffold for further clinical trials.

Local tissue rearrangement has limitations in that there may be areas that are difficult to reconstruct, such as an upper inner quadrant defect, the learning curve effect, and the possibility of cancer recurrence. According to the locally advanced breast cancer treatment algorithm that was presented by Peled et al. [21], only breast conserving surgery was performed when small resection was planned in patients with small breast with minimal ptosis. In this case, if the PCL scaffold is used, the reconstruction can be completed by immediately inserting the defect after partial mastectomy regardless of those considerations. As the PCL scaffold is hydrolyzed with time, capsular contracture can be reduced and it is considered safe to be inserted into any plane, such as subpectoral or prepectoral [22].

Also, the PCL scaffold can be utilized for radiotherapy. As radiotherapy is accompanied by breast conserving surgery, considering the research results that the removal rate of implants was high after receiving radiation therapy after reconstruction using implants [23], it is necessary to further study the changes in the PCL scaffold for radiation therapy. It should also be considered that chemotherapy or hormone therapy may cause tissue complications [24]. It is also important to ensure oncologic safety. Further research is needed to determine how it will be related to patient satisfaction if applied to actual clinical practice.

## 5. Conclusions

The PCL scaffold was manufactured with a sphere structure for a partial mastectomy. The PCL scaffold showed the high compressive strength and morphology recovery properties. The implantation of the PCL scaffold in a rat partial mastectomy model increased adipose tissue restoration and fibrous tissue was reassembled in the PCL scaffold with collagen when a little of the soft tissue was restored by PCL scaffold with rat breast fragment. PCL scaffold implantation after a partial mastectomy somewhat increased the pro-inflammatory cytokine and microcalcification. The implantation of the PCL scaffold with collagen and breast fragments showed no significant difference of inflammatory response. The restoration of adipose tissue was poorly observed compared to PCL scaffold implantation but fibrous tissue was fully restored by implantation of the PCL scaffold with collagen. Thus, the present study could provide an alternative strategy for the restoration of soft tissue depending on the tissue characteristics.

## Data Availability

All data in the present study are included in the published article.

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
