# Peer review of "Breast Tissue Restoration after the Partial Mastectomy Using Polycaprolactone Scaffold"

_polymers, 2022, doi:10.3390/polym14183817_

Round 1

Reviewer 1 Report

The manuscript reported by Jwa et al entitled as “Breast tissue restoration after the partial mastectomy using Polycaprolactone scaffold” attempts to explore the capability of 3D printed PCL scaffolds for breast tissue restoration during mastectomy. The idea of using a biocompatible polymer like Polycaprolactone do have relevance and can contribute to the breast cancer treatment. The systematic biological results presented in the study is convincing. However, the manuscript still needs further improvements before the acceptance. Hence, in my opinion the manuscript can be accepted for publication after major revisions. My specific comments about the manuscript are as follows.

1.      Authors are testing 3 different PCL scaffolds such as PCL scaffold, PCL-collagen scaffold, and the PCL scaffold with breast tissue fragment. However, the details regarding the fabrication of PCL-collagen scaffold, and the PCL scaffold with breast tissue fragment cannot be found on the manuscript. Authors should include this information in the materials and methods section.

2.      In Fig 1B, authors have only compared the compressive strength of the PCL scaffolds only. They should also include the compressive strength analysis of PCL-collagen scaffold, and the PCL scaffold with breast tissue fragment. This will be helpful to compare the mechanical property differences between the different scaffold’s groups tested.

3.      Authors should perform Scanning Electron Microscopy (SEM) analysis of the 3D printed scaffold. This will provide more detailed information on the pore architecture and connectivity of the pores of the 3D printed scaffold.

4.      The discussion section seems to be too short with out explaining the results presented in the manuscript. Authors should try to include more detailed  comparative evaluation of the 3 scaffolds systems which they have used for the testing. Also, they should clearly state which system is better and justify the reason with some potential mechanistic insights. This information will make the manuscript more robust.

5.      Please include a conclusion section to the manuscript and summarize the important findings of the present study.

……………………………………………………………………………………………………………………………………………………..

Author Response

The manuscript reported by Jwa et al entitled as “Breast tissue restoration after the partial mastectomy using Polycaprolactone scaffold” attempts to explore the capability of 3D printed PCL scaffolds for breast tissue restoration during mastectomy. The idea of using a biocompatible polymer like Polycaprolactone do have relevance and can contribute to the breast cancer treatment. The systematic biological results presented in the study is convincing. However, the manuscript still needs further improvements before the acceptance. Hence, in my opinion the manuscript can be accepted for publication after major revisions. My specific comments about the manuscript are as follows.

  1. Authors are testing 3 different PCL scaffolds such as PCL scaffold, PCL-collagen scaffold, and the PCL scaffold with breast tissue fragment. However, the details regarding the fabrication of PCL-collagen scaffold, and the PCL scaffold with breast tissue fragment cannot be found on the manuscript. Authors should include this information in the materials and methods section.

Thank you for your comment. We fabricated the PCL scaffold first, and then provided the PCL-col and PCL-tissue group during surgery immediately. We fixed the manuscript in Materials and Methods 2.3 followed by your kind suggestion as below:

“To investigate the tissue restoration capacity of PCL scaffold and the synergistic effects of PCL scaffold with extracellular matrix and autologous tissue implantation after partial mastectomy, 3 types of PCL scaffold were examined: 1) PCL scaffold only (PCL); 2) PCL scaffold with injection of 1 ml of 3% collagen type 1 solution within PCL scaffold (MSbio, Inc., Seongnam, Korea) (PCL-col); 3) PCL scaffold with autologous mammary gland fragment produced by surgical procedure immediately (PCL-tissue).”

  1. In Fig 1B, authors have only compared the compressive strength of the PCL scaffolds only. They should also include the compressive strength analysis of PCL-collagen scaffold, and the PCL scaffold with breast tissue fragment. This will be helpful to compare the mechanical property differences between the different scaffold’s groups tested.

Thank you for your comment. We measured the mechanical properties such as compressive strength and shape recovery of PCL scaffolds but not PCL collagen scaffold and PCL scaffold with breast tissue fragment. These groups were provided for the test of multiple utilization of PCL scaffold. We focus on the characteristics of PCL scaffold and another 2 types of scaffold was manufactured during surgery immediately and expected that mechanical strength of collagen solution and breast fragment are definitely lower so that these materials cannot affect the mechanical strength of PCL scaffold. Please take these circumstances into account.

  1. Authors should perform Scanning Electron Microscopy (SEM) analysis of the 3D printed scaffold. This will provide more detailed information on the pore architecture and connectivity of the pores of the 3D printed scaffold.

Thank you for your comment. We provide the SEM analysis of the PCL scaffold in the Figure 1B and Results 3.1, line 4 followed by your kind suggestion as below:

“To evaluated the surface structure of PCL scaffold, SEM image was obtained (Figure 1B). SEM image showed that PCL scaffold has a multi-layer structure to sustain the sphere structure and PCL scaffold is well-defined without nanoscale pore distribution.”

  1. The discussion section seems to be too short without explaining the results presented in the manuscript. Authors should try to include more detailed comparative evaluation of the 3 scaffolds systems which they have used for the testing. Also, they should clearly state which system is better and justify the reason with some potential mechanistic insights. This information will make the manuscript more robust.

Thank you for your comment. We included the discussion of explanation of our PCL scaffold and analysis with it in the 1st~3rd paragraph as your kind suggestion.

  1. Please include a conclusion section to the manuscript and summarize the important findings of the present study.

Thank you for your comment. We provide the conclusion section in the manuscript followed by your kind suggestion as below:

“The PCL scaffold was manufactured with sphere structure for partial mastectomy. PCL scaffold showed the high compressive strength and morphology recovery properties. Implantation of PCL scaffold in rat partial mastectomy model increased adipose tissue restoration and fibrous tissue was reassembled in PCL scaffold with collagen when a little of soft tissue restored by PCL scaffold with rat breast fragment. PCL scaffold implantation after partial mastectomy somewhat increased the pro-inflammatory cytokine and micro-calcification. Implantation of PCL scaffold with collagen and breast fragment showed no significant difference of inflammatory response. Restoration of adipose tissue was poorly observed compared to PCL scaffold implantation but fibrous tissue was fully restored by implantation of PCL scaffold with collagen. Thus, the present study could provide an alternative strategy to restoration of soft tissue depending on the tissue characteristics.”

Reviewer 2 Report

Polymers-1867571

Title: Breast tissue restoration after the partial mastectomy using Polycaprolactone scaffold

OVERVIEW

The paper addresses the reconstruction of breast tissue using PCL and PCL associated with collagen and the breast tissue itself. The data is very interesting. I believe that the manuscript needs some adjustments for its publication. Next, I will make some comments.

INTRODUCTION

The Introduction is well written. The problem is well contextualized. I only suggest, at the discretion of the authors, two corrections of form:

- In the first line it says mastectomy twice in a very short sentence. It's not wrong, but to be more elegant I suggest changing the first “mastectomy” to “surgery”

- In the first paragraph, line 6, is the acronym BCS. Excuse me if I'm wrong, but would it be "breast cancer surgey"? It is not in the text. Correction is worth it.

MATERIAL AND METHODS

In general, the Material and Methods is well written. I understand that it is possible to reproduce the tests by the method described. I have just a few suggestions:

- In item 3.2 there are two variations of PCL (PCL-Col and PCL-Tissue). I suggest that these variations are described in item 2.1. How was collagen added? Was it just a coating or is there a chemical bond with the PCL? Is it a copolymer? A blend? It is also worth mentioning how the breast tissue in PCL-tissue was extracted. Was there tissue culture? Were they entered into the PCL and deployed immediately? It is worth clarifying these points.

RESULTS

Regarding the Results, most of them are very interesting. Congratulations to the authors. I have some comments:

- 3.1. About “Mechanical properties of PCL scaffold” are the reductions observed in sterile and non-sterile material significant? I understand not. This is important information. I suggest that it be stated explicitly.

- 3.2. Very interesting data from the “Tissue restoration after PCL scaffold implantation”. I ask if there are statistical variations between the groups, for example: PCL, PCL-Col and PCL-Tissue are different at 4 weeks. Are these numbers significant? The same is true at 6 weeks. As the data is relevant, it is worth this clarification.

- The data presented in 3.3 and 3.4 are clear to me.

DISCUSSION

About the Discussion, the work has such interesting results that I believe that the discussion could be better explored. I would like at least point to be discussed.

- The authors say on page 5 (item 3.3): “Thus, it was determined that fat tissue was restored within PCL scaffold and fibrous tissue was restored by collagen and breast tissue fragment co-implantation”. I ask, in the experience of surgeons, what the authors understand as most suitable for breast tissue reconstruction.

REFERENCES

References must be suitable for the Polymers format.

References listed up to 25 also appear. Is there reference 26? In the text is cited up to 18 in the discussion. Was something lost?

Author Response

The paper addresses the reconstruction of breast tissue using PCL and PCL associated with collagen and the breast tissue itself. The data is very interesting. I believe that the manuscript needs some adjustments for its publication. Next, I will make some comments.

INTRODUCTION

The Introduction is well written. The problem is well contextualized. I only suggest, at the discretion of the authors, two corrections of form:

- In the first line it says mastectomy twice in a very short sentence. It's not wrong, but to be more elegant I suggest changing the first “mastectomy” to “surgery”

Thank you for your comment. We changed the first world “mastectomy” to “surgery” and described it as “Mastectomy for breast cancer started with radical surgery” in the first paragraph, line 1 followed by your kind suggestion.

- In the first paragraph, line 6, is the acronym BCS. Excuse me if I'm wrong, but would it be "breast cancer surgey"? It is not in the text. Correction is worth it.

Thank you for your comment. We changed the manuscript and described it as “A recent study found that up to 80% of breast cancers were safely treated with breast conservative surgery (BCS).” in the first paragraph, line 6 followed by your kind suggestion.

MATERIAL AND METHODS

In general, the Material and Methods is well written. I understand that it is possible to reproduce the tests by the method described. I have just a few suggestions:

- In item 3.2 there are two variations of PCL (PCL-Col and PCL-Tissue). I suggest that these variations are described in item 2.1. How was collagen added? Was it just a coating or is there a chemical bond with the PCL? Is it a copolymer? A blend? It is also worth mentioning how the breast tissue in PCL-tissue was extracted. Was there tissue culture? Were they entered into the PCL and deployed immediately? It is worth clarifying these points.

Thank you for your comment. We fabricated the PCL scaffold first, and then produced the PCL-col as injection of 1 ml of 3% collagen solution within PCL scaffold and the PCL-tissue as implantation of autologous breast tissue within PCL scaffold during surgical procedure immediately. We changed the paragraph in item 2.3 as “To investigate the tissue restoration capacity of PCL scaffold and the synergistic effects of PCL scaffold with extracellular matrix and autologous tissue implantation after partial mastectomy, 3 types of PCL scaffold were examined: 1) PCL scaffold only (PCL); 2) PCL scaffold with injection of 1 ml of 3% collagen type 1 solution within PCL scaffold (MSbio, Inc., Seongnam, Korea) (PCL-col); 3) PCL scaffold with autologous mammary gland fragment produced by surgical procedure immediately (PCL-tissue).”

RESULTS

Regarding the Results, most of them are very interesting. Congratulations to the authors. I have some comments:

- 3.1. About “Mechanical properties of PCL scaffold” are the reductions observed in sterile and non-sterile material significant? I understand not. This is important information. I suggest that it be stated explicitly.

Thank you for your comment. We measured the compressive strength of PCL scaffold before and after E-beam sterilization. After sterilization, the compressive strength of PCL scaffold was significantly decreased. We described it in Results 3.1, line 14. As below:

“Thus, compressive strength of PCL scaffold was somewhat decreased after sterilization but the morphological recovery ability was maintained.”

- 3.2. Very interesting data from the “Tissue restoration after PCL scaffold implantation”. I ask if there are statistical variations between the groups, for example: PCL, PCL-Col and PCL-Tissue are different at 4 weeks. Are these numbers significant? The same is true at 6 weeks. As the data is relevant, it is worth this clarification.

Thank you for your comment. We confirmed the time-dependent tissue restoration within PCL scaffold using microCT imaging and manual quantification as shown Figure 2B and 2C. We sought to measure the breast tissue within PCL scaffold and compare the time-dependent breast tissue restoration in the same group, for example: 1) PCL 4 months to PCL 6 months, 2) PCL-col 4 months to PCL-col 6 months. In the Figure 3, the characteristics of tissues within PCL scaffold was different in each group so that we thought that comparison of soft tissue in the same date seems to be not important.

- The data presented in 3.3 and 3.4 are clear to me.

DISCUSSION

About the Discussion, the work has such interesting results that I believe that the discussion could be better explored. I would like at least point to be discussed.

- The authors say on page 5 (item 3.3): “Thus, it was determined that fat tissue was restored within PCL scaffold and fibrous tissue was restored by collagen and breast tissue fragment co-implantation”. I ask, in the experience of surgeons, what the authors understand as most suitable for breast tissue reconstruction.

Thank you for your comment. We guess that PCL ball single implantation would be suitable for breast reconstruction and PCL scaffold with collagen would be effective for other fibrous tissue reconstruction such as cartilage and bone. We changed the manuscript in 1st and 2nd paragraph as below:

REFERENCES

References must be suitable for the Polymers format.

References listed up to 25 also appear. Is there reference 26? In the text is cited up to 18 in the discussion. Was something lost?

Thank you for your comment. We changed the format and list of reference as your kind suggestion. Format of the present paper was fixed followed by MDPI style.